# Preparation and Characterization of PVDF–TiO_2_ Mixed-Matrix Membrane with PVP and PEG as Pore-Forming Agents for BSA Rejection

**DOI:** 10.3390/nano13061023

**Published:** 2023-03-12

**Authors:** Rianyza Gayatri, Ahmad Noor Syimir Fizal, Erna Yuliwati, Md Sohrab Hossain, Juhana Jaafar, Muzafar Zulkifli, Wirach Taweepreda, Ahmad Naim Ahmad Yahaya

**Affiliations:** 1Institute of Chemical and Bioengineering Technology, Universiti Kuala Lumpur Malaysian, Alor Gajah 78000, Melaka, Malaysia; rianyza.gayatri@s.unikl.edu.my (R.G.); syimir.fizal@s.unikl.edu.my (A.N.S.F.); muzafar@unikl.edu.my (M.Z.); 2Polymer Science Program, Division of Physical Science, Faculty of Science, Prince of Songkla University, Hat-Yai 90110, Songkhla, Thailand; wirach.t@psu.ac.th; 3Program Study of Chemical Engineering, Faculty of Engineering, Universitas Muhammadiyah Palembang, Jalan A. Yani 13 Ulu Kota, Palembang 30263, Indonesia; erna_yuliwati@um-palembang.ac.id; 4HICoE-Centre for Biofuel and Biochemical Research, Institute of Self-Sustainable Building, Department of Fundamental and Applied Sciences, Faculty of Science and Information Technology, Universiti Teknologi PETRONAS, Seri Iskandar 32610, Perak Darul Ridzuan, Malaysia; sohrab.hossain@utp.edu.my; 5Advanced Membrane Technology Research Centre (AMTEC), Faculty of Chemical and Energy Engineering, Universiti Teknologi Malaysia, Skudai 81310, Johor, Malaysia; juhana@petroleum.utm.my

**Keywords:** mixed-matrix membranes, PVDF, PVP, PEG, TiO_2_, hydrophilic modification, ultrafiltration, phase inversion, BSA rejection

## Abstract

Polymeric membranes offer straightforward modification methods that make industry scaling affordable and easy; however, these materials are hydrophobic, prone to fouling, and vulnerable to extreme operating conditions. Various attempts were made in this study to fix the challenges in using polymeric membranes and create mixed-matrix membrane (MMMs) with improved properties and hydrophilicity by adding titanium dioxide (TiO_2_) and pore-forming agents to hydrophobic polyvinylidene fluoride (PVDF). The PVDF mixed-matrix ultrafiltration membranes in this study were made using the non-solvent phase inversion approach which is a simple and effective method for increasing the hydrophilic nature of membranes. Polyvinylpyrrolidone (PVP) and polyethylene glycol (PEG) as pore-forming chemicals were created. Pure water flux, BSA flux, and BSA rejection were calculated to evaluate the mixed-matrix membrane’s efficiency. Bovine serum albumin (BSA) solution was employed in this study to examine the protein rejection ability. Increases in hydrophilicity, viscosity, and flux in pure water and BSA solution were achieved using PVP and PEG additives. The PVDF membrane’s hydrophilicity was raised with the addition of TiO_2_, showing an increased contact angle to 71°. The results show that the PVDF–PVP–TiO_2_ membrane achieved its optimum water flux of 97 L/(m^2^h) while the PVDF–PEG–TiO_2_ membrane rejected BSA at a rate greater than 97%. The findings demonstrate that use of a support or additive improved filtration performance compared to a pristine polymeric membrane by increasing its hydrophilicity.

## 1. Introduction

Water pollution has become the leading cause of environmental deaths worldwide [1]. Around 1.2 billion people lack access to safe drinking water and millions die each year from diseases transmitted by contaminated water or human waste [2]. Reverse osmosis, distillation, chlorination, and coagulation are just a few of the technologies employed to filter water, but none of them can eliminate contaminants at the nanoscale. Traditional filtering mediums perform poorly for nanometer-sized environmental remediation despite their excellent filtration efficiency [3]. Researchers are focusing on a relatively advance filtration method known as membrane filtration which is known to possessed several benefits including lower power consumption, chemical-free operation, scalability, and the ability to operate at lower temperatures [4]. Membranes are selectively permeable materials that permit some molecules to pass while blocking others depending on the size of the molecules and the pore diameter [5].

The creation of polymeric materials and polymeric separation membranes has emerged as the most recent area of interest [6,7]. Polymeric membranes provide simple fabrication methods for a wide range of pore diameters and simple modification procedures that enable low-cost and easy commercial scaling. These materials are prone to fouling and are sensitive to extreme working conditions such as pH, temperature, and pressure. Polyvinylidene fluoride (PVDF) is an excellent example of a polymer used extensively in membrane fabrication due to its high thermal and mechanical strength [8]. PVDF is an effective membrane material for producing microfiltration and ultrafiltration membranes because of its superior chemical resistance, robust thermal resistance, ease of fabrication, and use in numerous industrial applications [9,10]. PVDF is a well-known thermoplastic substance with a high melting point and excellent crystallinity [11]. The inherent hydrophobicity of PVDF limits its application [12]. The resulting membrane fouling and membrane densification due to the materials’ inherent hydrophobicity have become urgent issues [13]. Many studies have been carried out to enhance the hydrophilicity of PVDF along with chemical grafting and plasma modification to expose hydrophilic groups to the PVDF surface of the membrane [14]. The simple inclusion of hydrophilic polymeric materials and nanoparticles into the membrane surface and bulk is made possible by blending modification [15]. Blending modification transforms membranes by interacting with compatible hydrophilic additives and polymers [16]. Mixed-matrix membranes (MMMs) are the solution to this challenge [17]. The rough and hydrophilic surface of the nanocomposite membranes was created by combining nanoparticles with polymers to improve the properties of polymer membranes (such as mechanical properties, anti-fouling properties, and antibacterial properties) [13].

Organic and inorganic nanomaterials have frequently been combined to imbue membranes with desirable functional properties [18]. A traditional PVDF casting membrane is prepared using the phase inversion method and mixing it with the hydrophilic polymer material is a simple and effective way to enhance the hydrophilic nature of the membranes [19]. Inorganic additives such as silica (Si), silica oxide (SiO_2_), zirconium dioxide (ZrO_2_), or titanium dioxide (TiO_2_) particles have been investigated for incorporation into membrane matrix to modify polymeric PVDF membrane surfaces [17,20,21]. Hydrophilic polymers, for example cellulose acetate phthalate (CAP), polyvinyl pyrrolidone (PVP), and polyvinyl alcohol (PVA), have also been studied to modify PVDF membranes [18,19]. By incorporating nanoparticles (NPs) in the fabrication of nano-composite membranes, nanotechnology advancements have broadened avenues of membrane modification [22]. Incorporating inorganic materials into organic polymer matrices has piqued the interest of many researchers due to its simplicity, mild conditions, and stable performance [15]. Fouling mitigation benefits of NP-based membranes, such as self-cleaning or anti-fouling properties, have been frequently covered [19,23]. This is primarily due to the hydrophilic enhancement of the membrane by the incorporated NPs. NP-related membrane functionality depends on NP dispersibility in membrane matrices [24]. Nanoparticles are frequently utilized to modify the surface of PVDF polymers due to their numerous applications [25]. The considerable changes in membrane properties found as a result of the dual behavior of the two components and interfacial interactions between the pure organic polymers and inorganic nanomaterials have drawn attention to this research on MMMs [26].

Titanium dioxide is a member of the transition metal oxide family, as is titanium (IV) oxide. TiO_2_ is a versatile material with many applications including in protective surface coatings, sensors, paints, water treatment, cosmetics, solar cells, batteries, and many others [24]. High chemical stability, low production costs, non-toxicity, and biodegradability are just a few of the many impressive features of titanium dioxide [27]. TiO_2_ nanoparticles are ideal for incorporating into a polymeric matrix for a hydrophilic filter because of their strong reactivity and high surface-to-volume ratio [28]. The hydrophilic and antifouling properties of the membrane can be improved by adding TiO_2_ NPs to the PVDF matrix [29]. Polar Ti-O bonds in TiO_2_ nanoparticles give them a high level of surface activity and the ability to act as adsorption carriers. TiO_2_ will polarize and produce a lot of hydroxyl groups due to ionization after absorbing water [13]. The hydrophilic, self-cleaning, antifouling, and antibacterial properties of the polymer membrane might thus be improved by adding TiO_2_ NPs [29]. Hydrophilicity and performance of PVDF membranes can also be enhanced by adding high molecular weight organics like PVP or PEG or inorganic additives like nanoparticles to the casting solution [30]. PVP and PEG may function as pore-forming agents to increase membrane hydrophilicity and pore interconnectivity [31].

Numerous studies have been conducted to investigate the performance of MMMs. Incorporating a second polymer, such as PVP, into the PES solutions resulted in highly porous membranes with well-connected pores and desirable surface properties. PVP mixing has demonstrated the ability to hydrophilize and adjust the porosity of the PES membrane [32]. Enayatzadeh et al. [30] used phase inversion to create blended flat sheet PVDF-based membranes with varying PVP and TiO_2_ ratios. They also reported a hydrophilic modification of PVDF membranes that reduced fouling after TiO_2_ addition. Adding TiO_2_ nanoparticles increased the viscosity of the casting solution, resulting in a thicker skin layer due to the delayed membrane formation mechanism. According to Lu et al. [33], incorporating GO-NBA into PVDF flat sheet membranes improves mechanical properties over the addition of GO due to better interfacial interaction, dispersity, and a more uniform crystal structure.

Only a few studies have considered and thoroughly discussed the effects of TiO_2_, PVP, and PEG incorporation into PVDF membranes. This study aims to create and compare the properties and the performance of PVDF/TiO_2_ mixed-matrix membranes with pore-forming agents like PVP and PEG. Significant efforts have been made in this study by adding TiO_2_ and pore-forming agents to hydrophobic PVDF polymeric matrix to address the issues in polymeric membranes and create membranes with improved hydrophilicity and properties. A novel systematic study has been carried out into the creation, characterization, and performance evaluation of modified membranes PVDF–PVP–TiO_2_ and PVDF–PEG–TiO_2_. Porosity and other obtained results were correlated for potential applications and testing in measuring pure water flux and Bovine Serum Albumin (BSA) rejection after preparation and characterization. The significance of this study is also to examine the impact of TiO_2_ and pore-forming chemical (PVP and PEG) addition on membrane properties and performances (pure water flux and BSA rejection). This study aims to show that adding PVP or PEG to a PVDF–TiO_2_ mixed-matrix membrane significantly affects its surface hydrophilicity, pore size, porosity, and surface morphology. Numerous studies have examined how to alter the surfaces or blends of PVDF membranes to tailor their performance. Investigation of other parameters, such as PVDF, TiO_2_, and additive concentrations (PVP and PEG), might be the subject of future study. The possibility of using mixed-matrix membranes to treat actual wastewater and detect BSA rejection is another potential future use.

## 2. Materials and Methods

### 2.1. Materials

Polyvinylidene Fluoride (PVDF) Kynar ^®^740 pellets as polymer-based membranes were obtained from Arkema Inc. Philadelphia, PA, USA. Merck, Rahway, NJ, USA, supplied N, N-dimethylacetamide (DMAc, >99%) as a solvent. Evonik GmbH, Essen, Germany, delivered TiO_2_ nanoparticles (Degussa P25 with average particle size ~21 nm). Sigma-Aldrich, St. Louis, MI, USA, provided polyethylene glycol (PEG) (MW = 10,000 g/mol) and polyvinylpyrrolidone (PVP) (40,000 g/mol) additive membranes. Sigma Aldrich supplied Bovine Serum Albumin (BSA) (MW = 66,000 g/mol). 

### 2.2. Methods

#### Membrane Fabrication

The flat sheet membrane was prepared using non-solvent phase inversion method (NIPS). A portion of PVDF pellets were pre-dried (24 h at 50 °C) and the dope solution was made by dissolving the PVP or PEG as an additive to the DMAc solvent before dispersing the appropriate amount of TiO_2_ into the solution. The PVDF (16 g) was dissolved in the solution which was then stirred at 450 rpm for 24 h at 60 °C. The residual air bubbles in the dope solution were removed by placing the mixture on an ultrasonicator for 30 min after it had completely dissolved. The membrane dope solution was left for 24 h before casting and the membrane was soaked in water for an entire day after casting. The membrane dried at room temperature for 24 h. The pure water flux, permeate flux, and BSA rejection were measured to determine the membrane’s performance. Table 1 shows the dope solution’s composition and the amount of TiO_2_ added.

### 2.3. Membrane Characterizations

Scanning electron microscopy (SEM) was applied to examine the membrane morphology at various magnifications. The SEM results also gave information regarding the pore diameter of each membrane and the membrane thickness. FTIR can display the change in the functional groups and elements in the polymers. Contact Angle Analysis was also employed to assess membrane hydrophilicity. The static contact angle of the membrane was evaluated using the sessile drop method and a Drop Meter A-100 contact angle system to characterize its wetting behavior. The Modular Compact Rheometer (MCR 302) was used to assess the viscosity of the dope solution at room temperature.

A LLOYD-LR30KPlus instrument was used for the tensile test. Tensile properties were evaluated by calculating values for tensile stress (in MPa), elongation at break (in percent), and elastic modulus (in MPa) in the NEXYGENPlus program. An average value was reported for each membrane that was calculated from three separate samples. The working conditions were as follows: the specimen’s length in the gauge section was 100 mm, the stroke speed was 50 mm/min, and the width was 20 mm.

The porosity and average pore radius of asymmetric porous membranes were determined. Membrane porosity was determined according to its dry/wet weights [34]. The following equation was used to calculate membrane porosity (1) [35]:(1)$$\in =\frac{\frac{\left(w1-w2\right)}{\rho w}}{\frac{\left(w1-w2\right)}{\rho w}+\frac{\left(w2\right)\;}{\rho p}}\times 100$$
where ∈ is the porosity of membrane (%), *w*_1_ is the wet membrane’s weight (g), *w*_2_ is the dry membrane’s weight (g), *p* is the polymer’s density (g/cm^3^), and *w* is the water density (g/cm^3^). Five cuts of 2 cm^2^ flat sheet membranes were chosen to prepare the wet and dry membranes after soaking in water for a day. The remaining water on the inner surface of the membranes was removed prior to weighing. Wet membranes were weighed after drying in a vacuum oven at 50 °C for 12 h.

### 2.4. Experiment with Filtration (Permeation Flux and Rejection Measurement)

A membrane’s performance is typically determined by its selectivity and flux rate. The volumetric liquid flow per unit area per unit time through a membrane is defined as flux [36]. Ultrafiltration experimental equipment was used to measure the permeation flux and the membrane rejection, as shown in Figure 1. Separation performance was evaluated by using a flat sheet membrane with a working pressure of 1.0 bar in the fluid membrane permeation ultrafiltration testing unit. The filtration area of a flat sheet membrane is 0.4299 cm^2^.

The membranes’ performance was assessed by measuring pure water flux using reverse osmosis (RO) water. Membrane rejection was measured using a feed of 1000 ppm Bovine Serum Albumin (BSA). The permeated solution was measured every 15 min for one hour to calculate the permeate flux and kept in a vial for permeate concentration measurement with a spectrophotometer for rejection percentage calculation. Water flux (J) of a membrane was calculated using Equation (2) [37]:(2)$$J=\frac{Q}{A\times t}$$
where *J* is the water flux (L/m^2^h), *Q* is permeate volume (L), *t* is time (h) required to obtain *Q*, and *A* is effective membrane area (m^2^).

The following equation was then used to determine membrane rejection [37]:(3)$$R=1-\frac{Cp}{Cf}\times 100$$
where R is the rejection percentage (%) and *C_p_* and *C_f_* are the BSA concentrations in the permeate (ppm) and feed (ppm), respectively. The concentration of BSA in permeate and feed were determined using a UV-vis spectrophotometer at the maximum absorption wavelength of 278 nm.

## 3. Result and Discussion

### 3.1. Membrane Characterizations

#### 3.1.1. Morphological Studies of PVDF Mixed-Matrix Membranes

The membranes’ morphology was studied using a scanning electron microscope (SEM) to represent the cross-sectional and surface of the membranes at various magnifications. Figure 2 shows SEM micrographs of the prepared PVDF UF membranes’ cross-sectional morphology at magnifications of 8.00 k. All prepared membranes had asymmetric morphology with a finger-like structure on top and a sponge-like structure on the bottom. Figure 2a depicts a pristine PVDF with no additives showing a thin skin layer and an asymmetric structure with a finger-like structure on the top membrane. The SEM images showed the morphological changes and suppressed the finger-like macro voids on the surface of the PVDF membrane, revealing the successful incorporation of TiO_2_ nanoparticles, PVP, and PEG. The pure PVDF membrane surfaces had a larger pore size which is consistent with [19]. Noticeable changes in morphology were seen with the addition of inorganic TiO_2_ nanoparticles and different pore-forming agents (PVP and PEG) in which the inner and outer finger-like macro voids were suppressed (Figure 2b–f). The finger-like structure beneath the top layer and the entire membrane structure was made of 16 %.wt PVDF. There was also a transition from the macroporous to the asymmetric structure in the membrane cross-section. A sponge-like substructure and a thicker layer of membranes were formed at the bottom of the membrane. A small amount of TiO_2_ nanoparticles was added to improve membrane morphology. TiO_2_ nanoparticles have a high specific area and are hydrophilic which will impact mass transfer during the fabrication process [25,28].

Steric hindrance effects of the PEG on the PVDF membrane assisted in the homogeneous dispersion of TiO_2_ nanoparticles on the surface and sub-layers which facilitated the formation of a finger-like structure [18]. Significant PEG chains attached to the TiO_2_ nanoparticles would make the casting solution more viscous, preventing the free formation of a finger-like structure on the membrane [38]. The optimal TiO_2_/PEG dose ratio must be investigated further. Incorporation of PEG into the dope solution caused the formation of many pores on the membrane surface. The increased demixing rate at the interface caused by increasing the PEG content may result in the rapid collapsing of polymer chains and the formation of macro voids between collapsed chains [30,31].

Figure 2a–d show the PVDF membranes at different thicknesses based on different membrane additives where the thicknesses were 30.5 µm, 63 µm, 66.7 µm, and 95 µm, respectively. The thickness of the PVDF–PVP membrane examined in Figure 2e was approximately 151 µm. Macro voids and an apparent finger-like structure can be seen in Figure 2f. This phenomenon may be attributed to the PVDF–TiO_2_-PVP membrane which shows that PVP and TiO_2_ can cause clumping and aggregation and form more macro voids. The thickness of the membrane was approximately 94 µm.

Top surface SEM micrographs of PVDF–TiO_2_ MMMs are shown in Figure 3. Connected pores on the surface of all membranes were observed at magnifications of 1 k (see Figure 3). Identified TiO_2_ nanoparticle dispersion in the synthesized membranes was consistent with the theory postulated by [39] which states that the thermodynamic stability of nanoparticles in a polymeric liquid is strongly correlated with the ratio of the linear polymer’s radius of gyration (Rg) to the radius of the nanoparticles (Rp) [8].

Table 2 provides the pore size of the different membranes. The pore size of the modified membrane was more significant than that of the pristine PVDF membrane (91.3 nm) due to the presence of high hydrophilicity PVP and PEG additives in the membrane solution. The pore size of the PVDF–TiO_2_-PVP membrane was 148 nm followed by that of the PVDF–TiO_2_-PEG membrane (98 nm). The exchange rate between solvent and non-solvent can be improved by using a phase inversion technique during the membrane production process, leading to the development of pores with a larger finger-like structure. PVP as an additive has been shown to have similar effects on the morphology of PVDF membranes [40]. TiO_2_ nanoparticles which have a strong affinity for H_2_O molecules can enhance the pore size of membranes that were fabricated with a PVP or PEG co-polymer and TiO_2_ nanoparticles compared to membranes that did not contain TiO_2_ nanoparticles. TiO_2_ nanoparticles have a high water affinity caused by strong hydrogen bonding between the water and their surface hydroxyl groups [41]. Incorporating TiO_2_-containing membranes into the coagulation bath would likely increase non-solvent molecule diffusion into the pores of membranes [42]. The pore sizes of PVDF–TiO_2_ (142 nm) were found to be larger than those of PVDF–PVP (118 nm) and PVDF–PEG (111 nm).

#### 3.1.2. FTIR Spectroscopy of Membranes 

PVDF membrane crystalline phases were identified via FTIR spectroscopy. Figure 4 displays the FTIR spectra of pure PVDF and PVDF membranes modified with various additives. The crystal phase of PVDF and the connection between the polymer molecules and the nanoparticles were established through FTIR analyses. The membrane curves in Figure 4 smoothly superposed at the higher wavenumbers and diverged at the lower wavenumbers and this difference was traced back to the incorporation of titanium dioxide [34]. The 1179 cm^−^^1^ absorption peak was identified as being caused by the stretching vibration of CF_2_ groups [43]. The deformed vibration of CH_2_ groups first showed up at a frequency of 1400 cm^−^^1^ [44]. Peaks at 760 cm^−^^1^ were characteristic of phase PVDF crystals [44,45]. PVDF’s crystal phase mainly manifested such that the crystalline phase of PVDF was unaffected by the addition of nanoparticles during the phase inversion process [34].

Peaks at 1400 cm^−1^ were linked to the deformation vibration of -CH_2_; 1274 cm^−1^ and 1179 cm^−1^ to the symmetrical and asymmetrical stretching of -CF_2_; 877 cm^−1^ to one of the characteristic peaks of PVDF; and 840 cm^−1^ to the stretching vibration of -CH [46]. According to [47], the peak at 1065–1070 cm^−1^ represents the -OH stretching vibration. Crystal forms in the PVDF–PEG membrane were indicated by an absorption band at 840 cm^−1^ as reported by [48]. A stretching vibrational peak of CO at 1650 cm^−1^ was observed for both PVDF–TiO_2_-PVP and PVDF membranes, confirming the presence of residues [10]. Studies have shown that PVP with a high molecular weight (MW > 10,000 g/mol) is more likely to be ensconced in the membrane [10]. The characteristic spectral peak associated with PVP was still discernible and the nanofiber membrane’s resistance to water may be enhanced by the residual PVP which functions as a binder within the fiber [13]. Spectra of PVP showed three distinct peaks at 1290 cm^−1^, 1660 cm^−1^, and 1463 cm^−1^, which can be attributed to stretching vibrations of C-N, C = O, and CH_2_ bonds, respectively [49].

#### 3.1.3. Contact Angle

The hydrophilicity of the membrane surfaces was evaluated using the sessile drop method and water contact angle. Superior hydrophilicity is indicated by a smaller contact angle with water [43]. Capillary absorption and wetting cause the contact angle of water to decay slowly over time on solid surfaces and the hydrophilicity of the membranes significantly impacts this process [8,41]. Wettability, porosity, pore size, surface roughness, and pore size distribution are theoretical determinants of the contact angle of a membrane [50]. Higher surface roughness is predicted to result in a larger contact angle when comparing membranes with similar hydrophilicities [51].

Contact angles of PVDF membranes and PVDF–TiO_2_ mixed-matrix membranes formulated with polyvinylpyrrolidone (PVP) and polyethylene glycol (PEG) are shown in Figure 5. The pure PVDF membrane had the largest contact angle while the value varies between the various mixed-matrix membranes. The fact that the contact angle of pure PVDF measured in this study was close to 96.5° demonstrates that this material is hydrophobic. The contact angle dropped to 73° after TiO_2_ was added. The PVDF–TiO_2_ membrane’s low contact angle compared to the others demonstrates that TiO_2_ can boost the hydrophilicity of the membranes. The contact angle was raised to about 83° and 80° after pore-forming chemicals like PEG and PVP were included. Hydrophilicity variations depend on TiO_2_ nanoparticle dispersion in the PVDF matrix, and PVDF–TiO_2_ membranes had the lowest contact angle [18]. The mixed-matrix membranes with pore forming agents had greater contact angles than the PVDF–TiO_2_ membrane: the contact angle of PVDF–TiO_2_-PVP was 82.568° and PVDF–TiO_2_-PEG was 80.046°.

The increased surface roughness due to the nano-TiO_2_ loading amount may be responsible for the elevated contact angle of these membranes [52]. The contact angles of the PVDF–PEG, PVDF–PVP, PVDF–PVP–TiO_2_, and PVDF–PEG–TiO_2_ membranes were smaller than that of the pristine PVDF membrane. TiO_2_ nanoparticles have a much higher affinity for water due to the abundant hydroxyl groups on their surfaces which also increase the membranes’ surface hydrophilicity [18]. TiO_2_ nanoparticles’ high water affinity due to the extensive hydrogen bonding between water and their surface hydroxyl groups is beneficial in modifying the hydrophilicity of membranes [41]. The dispersion, hydrophilicity, and self-cleaning properties of TiO_2_ nanoparticles in the membrane are all negatively impacted by their aggregation from their initial size typically around 20 nm to several hundreds of nanometers due to their high surface energy [18]. Previous studies found that incorporating 0.156% TiO_2_ nanoparticles into composite membranes lowered contact angles by 2.8 percent, with PEG used as a dispersant of TiO_2_ nanoparticles before the casting solution due to its steric hindrance effects [21]. The distribution of nano-TiO_2_ may be hampered as the dosage of TiO_2_ nanoparticles increases because the dosed PEG can only offer steric hindrance effects for a limited number of nanoparticles [21]. PEG’s ability to create the hydration layer via hydrogen bonds that are relatively easy to break and reform during the immersing phase inversion process aids pore formation in composite membranes [53]. The nanoparticles must be finely dispersed so that a hydrated layer can form on the membrane surface which increases the surface area and the wettability of the membrane [8]. The results showed that PVDF–TiO_2_ was more hydrophilic than the pristine PVDF membrane while PVDF–PVP–TiO_2_ was less hydrophilic. This difference can be attributed to the increased clustering of TiO_2_ nanoparticles on the PVDF–PVP–TiO_2_ membrane surface which reduces surface wettability [34]. Hydrophilicity and permeability may be decreased due to the reduced rate of molecule exchange between the solvent and non-solvent [34].

#### 3.1.4. Membrane Porosity

Table 2 shows the calculated porosity of the fabricated mixed-matrix membranes. As can be seen, adding PVP to the PVDF membrane as an additive resulted in the highest porosity (88.28%) when compared to the pristine PVDF membrane (46.49%). The PVP additive sped up the solvent and non-solvent exchange rate during the whole phase inversion procedure, giving the PVP chain time to migrate onto the PVDF membrane surface and create a highly porous membrane [54]. PEG-modified membranes had a higher porosity than PVDF and PVDF–TiO_2_ membranes. The porosity of the modified membranes was higher than that of the pristine membrane because of the incorporation of PVP, PEG additives, and TiO_2_ nanoparticles [18]. The PVDF–TiO_2_ membrane had lower porosity than the PVDF–PEG–TiO_2_ and PVDF–PVP–TiO_2_ membranes due to TiO_2_ aggregation and blocked membrane pores [46]. Porosity increases and pore size decreases when only a small amount of additive is added to the polymer and this phenomenon is caused by the different hydrophilic and interaction properties of TiO_2_ and PVDF [19]. PVDF is a hydrophobic polymer, whereas TiO_2_ is a hydrophilic additive [55]. A certain amount of TiO_2_, PVP, and PEG increases the hydrophilicity and porosity of the membrane surface [13,18]. Water transport resistance increases despite the hydrophilic surface as pore size and porosity decrease and the dense top layer thickens. Adding 2% PEG to the PVDF–TiO_2_ membrane increased the percentage of surface porosity from 8% to 80%, proving that an increase in PEG content improves surface porosity as shown in Table 3. This result is consistent with microscopic research showing that PEG can create pores and holes on the surface of a membrane [43].

#### 3.1.5. Tensile Test

The membranes’ mechanical strength was also studied and the results are given in Table 4. PVDF had a tensile strength of 2.07 kPa and a tensile elongation at a break of 32.21 percent when unmodified but this value dropped to 1.15 kPa when TiO_2_ was added. The tensile strength of the pure PVDF membranes was greater than that of the PVDF–TiO_2_ membranes suggesting that the incorporation of TiO_2_ into the membrane matrix degraded this mechanical property [28]. Composite membranes containing TiO_2_ were more porous than those made from pure PVDF and had lower mechanical strength [28]. Young’s modulus of membrane substrates was reduced from 32.27 to 18.46 MPa [56].

The elastic modulus of the PVDF membrane was reduced further by incorporating PEG and PVP. The tensile strength was improved to 4.06 Kpa after the pore-forming agent PEG was added to the PVDF–TiO_2_ membrane surfaces. The elongation at the break of PVDF–TiO_2_ membranes was raised from 13.39 to 30.84% after PEG was added [57]. The mechanical properties of the polymer substrate were shown to be well preserved despite the addition of 2 wt% of PVP to the PVDF–TiO_2_ (the tensile strength decreased from 1.15 to 0.97 MPa) [56]. Higher PVP concentrations lead to decreased mechanical properties due to an increase in porosity (a longer finger structure) [58]. PVDF–PVP–TiO_2_ membranes had the lowest elastic modulus and tensile strength (4.47 Kpa and 0.97 Kpa, respectively) because of the addition of PVP. Pore size and pore structure are two morphological parameters that have been shown to have a major impact on the properties of polymeric membranes.

The nanoparticles’ reinforcing effect on mechanical strength may be canceled out by the weakening influence of porosity [28]. This overlap, however, may not always occur at a different concentration of TiO_2_. It has been reported that the predominance of crystallites contributed to the enhancement of the membrane’s mechanical properties [59], suggesting that the α crystalline form (Fα) of PVDF may also play a role in defining the membrane’s mechanical strength. The higher mechanical strength of PVDF membranes and membranes manufactured at higher temperatures may be explained by the increase in Fα with temperature and the higher in Fα pristine PVDF membranes [28]. It is widely recognized that the membrane’s mechanical properties are dependent on the polymer molecular weight and the polymer concentration in the solution with an increase in either parameter improving the membrane’s mechanical properties [60].

#### 3.1.6. Viscosity

Viscosity is a critical variable in the membrane formation process because of its role in phase separation kinetics [28]. The viscosity of various room-temperature membrane dope solutions with and without TiO_2_ and pore-forming agents is shown in Table 5. The lowest viscosity was observed in pure PVDF around 1577 MPa.s. The viscosity of a pristine membrane was increased when TiO_2_ nanoparticles were incorporated into it. This finding makes intuitive sense since the former is simply the dissolved polymer solution while the latter is a suspension of well-dispersed TiO_2_ nanoparticles [28].

The membrane’s viscosity was greatly augmented by incorporating pore-generating agents such as PVP and PEG. Consistent with prior research, we found that elevating the concentration of a water-soluble pore-forming chemical in the dope solution resulted in a more porous membranes [61]. Entanglement between the PVDF and pore-forming agent chains may contributed to the higher viscosity of the dope solutions [31]. Our results showed that PVDF–TiO_2_ membrane viscosity increased to 1789 MPa.s upon addition of TiO_2_ and further increased to 1992 MPa.s upon addition of PVP. PVDF–TiO_2_ membrane viscosity was slightly lowered when PEG was added (1719.8 MPa.s). The viscosity of the PVDF membranes was measured using both PEG and PVP alone as pore-forming agents: the PVDF–PVP membrane had the highest viscosity at 2917 MPa.s, while the PVDF–PEG membrane had a much lower value at 2007.6 MPa.s. It was hypothesized that the substantial PEG chains adhering to the TiO_2_ nanoparticles would enhance the viscosity of the casting solution which prevents the free creation of a finger-like structure for membrane insertion [38]. Increasing the concentration of the polymer PVP raises the viscosity of the precursor solution since the polymer chain of PVP is connected to the viscosity of the solution [62]. High dope solution viscosity has been observed to slow the exchange of solvent and non-solvent during the phase-inversion process, resulting in smaller surface pores [33]. Larger macro voids also can be reduced by increasing the viscosity of the fluid [63].

### 3.2. The Membrane Performance Analysis

#### 3.2.1. Water Permeation Test

The water permeation test data are presented as pure water flux using Reverse Osmosis (RO) water, and permeate flux using BSA as feed and BSA rejection test. Figure 6 depicts the pure water flux results of pristine PVDF and PVDF mixed-matrix membranes. The fabricated membranes’ hydrophilicity, porosity, and pore size were critical for flux measurements, and it should be noted that the reported values for each type of membrane were the average of four replicates using different membranes. The permeating fluxes showed that adding TiO_2_ and pore-forming agents increased the permeability of the mixed-matrix membranes [8]. Water permeate flux performance through the membrane showed that this is due to increased hydroxyl groups on the TiO_2_ surface which improves permeate water flux and increases porosity in the membrane structure despite its more hydrophobic properties [15].

The flux was 2.74 L/m^2^h for the pristine PVDF membrane and 7.56 L/m^2^h with TiO_2_ but the difference was not statistically significant due to the non-porous membrane. This study used PEG and PVP as pore-forming agents to make the membrane more porous. PVP works by increasing the membrane’s hydrophilicity [40]. PVP-modified membranes have higher water flux than PEG-modified membranes due to an increase in the thickness of the sponge-like structure on the support membrane which drastically alters the resistance to water transport [19]. PVDF–TiO_2_ with PVP produced the highest flux (97.01 L/m^2^h) while its BSA rejection was the lowest.

The current study showed that adding PVP to a PVDF–TiO_2_ membrane can improve the flux. In comparison to the other types of membranes, PVDF–TiO_2_ with PVP produced the highest pure water flux (97.01 L/m^2^h). The PVDF–TiO_2_-PEG membrane had a lower flux of 12.79 L/m^2^h due to TiO_2_ nanoparticle aggregation on the membrane’s surface, which blocked the membrane pores; some reports have also shown that PVP creates more pores than PEG [18,64].

Table 6 compares water fluxes between this current study and previous works. Mahdavi et al. [65] discovered that adding TiO_2_ nanoparticles to the membrane could significantly improve water flux; the PVDF-PVDFg-PVP-TiO_2_ membrane flux was 41.89 L/m^2^h. Teow et al. [8] demonstrated that PVDF–TiO_2_ MMMs with homogeneous TiO_2_ (X500) nanoparticle distribution had the highest membrane water permeability, implying that X500 (fully anatase) had the most hydrophilic behavior of the membranes tested. Water flux was calculated to be 58.81 ± 1.96 L/m^2^h. Teow et al. [52] discovered that mixed-matrix membranes always have higher fluxes than neat membranes, with a PVDF–TiO_2_ mixed-matrix membrane using X500 having a flux of 45.36 L/m^2^h. Ong et al. [37] investigated PVDF ultrafiltration membranes with varying molecular weights of TiO_2_ and PVP. Incorporating PVP into the PVDF–TiO_2_ membrane could improve its properties and performance. This study used PVP as a second agent to enhance the flat sheet PVDF membrane properties due to its high water flux (72.2 L/m^2^h) [37]. Based on a similar study [34], the pure water flux of PVDF with 0.5% TiO_2_ gave the highest value of 105.1 L/m^2^h. Pure water flux was directly related to the number of pores and pore size on the membrane surface (top layer porosity). According to Abba et al. [64], dispersing 1.0 wt% TiO_2_ into PVDF–PVP dope solution improved membrane performance regarding flux (223 L/m^2^h). Following a similar study [18], the PVDF–TiO_2_-PEG had a flux of 65.74 ± 1.77 L/m^2^h. Further study should analyze the other parameters like increasing the PVP or PEG concentration or use these pore-forming agents with other molecular weights to increase the flux.

#### 3.2.2. BSA Flux and Rejection Test 

Figure 7a,b displays the BSA solution flux and BSA rejection results for all produced membranes. Protein fouling, concentration polarization, and increased viscosity of the BSA solution decreased membrane flux compared to pure water [66]. There was a correlation between water flux and BSA flux, with the PVDF–TiO_2_-PVP membrane showing the highest flux (89.09 L/m^2^h) and the PVDF–TiO_2_-PEG membrane having the third lowest BSA flux (below 10 L/m^2^h). The second highest BSA flow was produced by the PVDF–PVP membrane (51.87 L/m^2^h). The PVP blending shift is not permanent, and a portion of the PVP gets washed away after use and PVP functions as a pore-forming agent [32]. Low rejection rates are caused by the membrane’s increased macro-void pores which lowers the BSA rejection [67]. There will be no surface area restrictions which will lead to low rejection. Increasing the membranes’ hydrophilicity on the surface primarily reduces the intensity of membrane blockage, leading to more rejection of the protein [67]. Hydrophilicity influences a membrane’s water permeability by increasing the contact between water molecules and pore walls [68]. Most protein retention was primarily controlled by membranes with dense outer surfaces that came into contact with protein solutions. A reduction in the hydrophobic contact between the hydrophilic membrane surface and the BSA protein may cause a slight increase in BSA rejection [43].

The membranes’ antifouling characteristics were greatly improved as seen by the increased hydrophilicity and improved rejection of BSA protein. The PVP-modified PVDF membranes had the lowest BSA rejection rates at only 0.75%, while PVDF–PEG–TiO_2_ had the highest BSA rejection at 98.66% similar to how higher flux resulted in reduced rejection. One study [21] has demonstrated that membranes can successfully reject BSA molecules due to their hydrophilic surface. The PVDF membrane’s pores were more susceptible to being clogged by BSA due to this reason. The dense surface also can simultaneously retain more BSA and have improved filtration resistance [69].

## 4. Conclusions

PVDF–TiO_2_ mixed-matrix membranes were created with higher hydrophilicity than pristine PVDF membranes due to the hydrophobic solid nature of PVDF membranes which makes them susceptible to protein fouling. Only a few studies have considered and thoroughly discussed the effects of TiO_2_, PVP, and PEG incorporation into PVDF membranes. This study made substantial attempts by adding TiO_2_ and pore-forming agents to a hydrophobic PVDF polymeric matrix to tackle concerns in polymeric membranes and create membranes with better hydrophilicity and properties. Mixed-matrix membranes including several additives, such as PVP, PEG, and nano-TiO_2_, were created utilizing the phase immersion process to examine their influence on PVDF membrane performance. This report reviewed the fabrication of flat sheet membranes and the impact of pore-forming and additional agents on membrane properties. A flat sheet membrane of 16%wt PVDF filled with TiO_2_ was successfully created with PVP weight concentrations of 2%wt which gave the highest pure water (97.01 L/m^2^h) and BSA flux (89.09 L/m^2^h). Scanning electron microscopy (SEM) revealed the asymmetrical structure of the membrane with finger-like structures on top and a sponge-like structure on the bottom. The measurement of contact angles demonstrated that numerous novel applications have been found for this material since TiO_2_ and PVP or PEG improved the membrane’s hydrophilicity. The adaptability of the PVDF mixed-matrix membrane was demonstrated by its use in a wide range of filtering processes including water to air. The addition of additives and pore-forming agents improved the hydrophilicity, porosity, viscosity, and mechanical strength of polymeric membranes.

## Figures and Tables

**Figure 1 nanomaterials-13-01023-f001:**
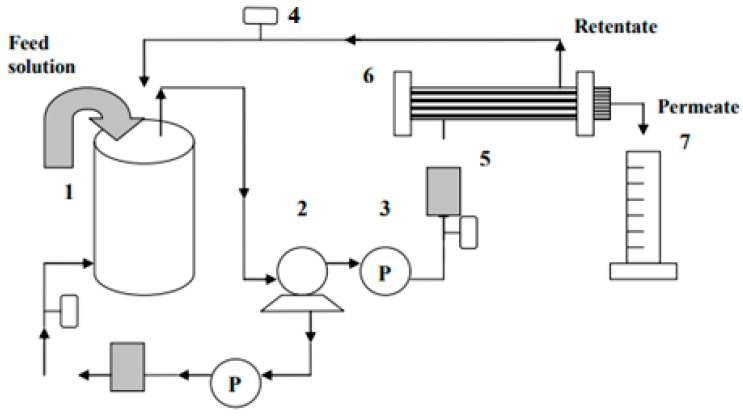
Ultrafiltration flat sheet membrane separation schematic diagram. (1) Feed tank; (2) pump; (3) pressure gauge; (4) control valve; (5) flow meter; (6) permeation test (flat sheet membrane module); (7) measuring cylinder.

**Figure 2 nanomaterials-13-01023-f002:**
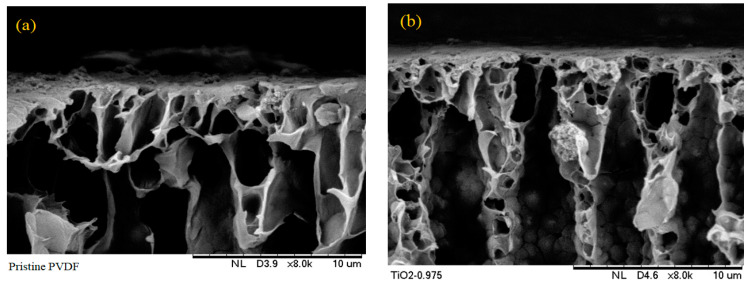
SEM images of membrane cross-sections: (**a**) pristine PVDF, (**b**) PVDF–TiO_2_, (**c**) PVDF–PEG, (**d**) PVDF–TiO_2_-PEG, (**e**) PVDF–PVP, (**f**) PVDF–TiO_2_-PVP.

**Figure 3 nanomaterials-13-01023-f003:**
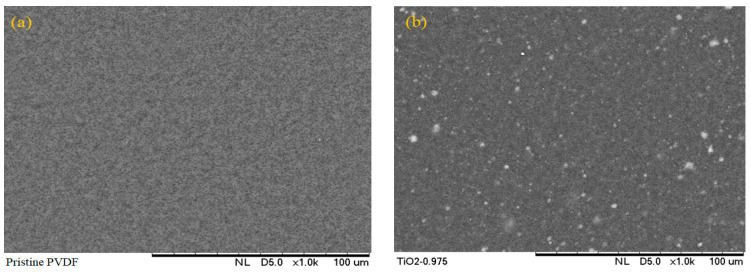
SEM micrographs the surface morphology of membranes: (**a**) pristine PVDF, (**b**) PVDF–TiO_2_, (**c**) PVDF–PEG, (**d**) PVDF–TiO_2_-PEG, (**e**) PVDF–PVP, (**f**) PVDF–TiO_2_-PVP.

**Figure 4 nanomaterials-13-01023-f004:**
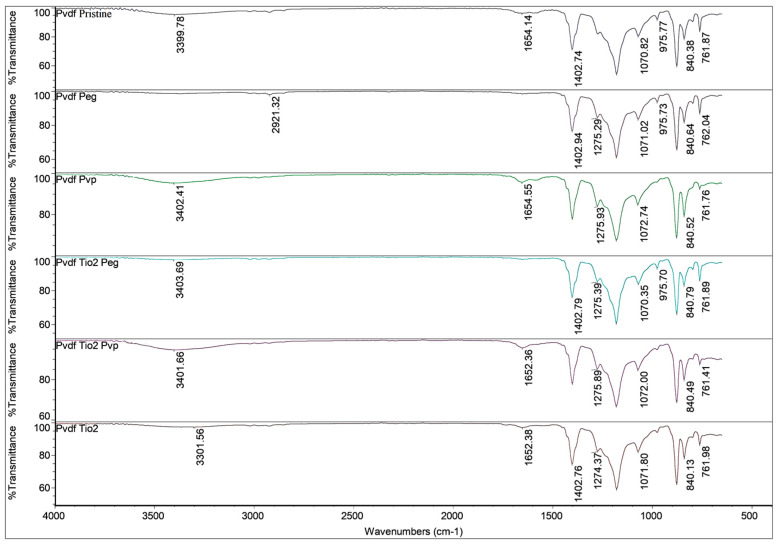
FTIR Spectra of Membranes.

**Figure 5 nanomaterials-13-01023-f005:**
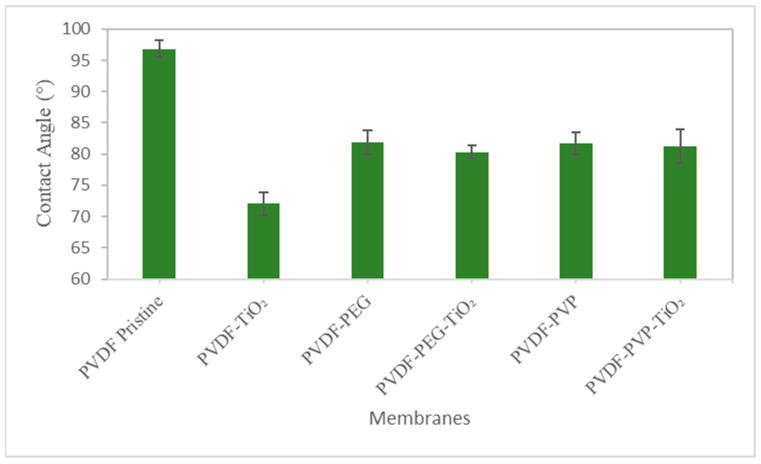
Contact angle of membranes.

**Figure 6 nanomaterials-13-01023-f006:**
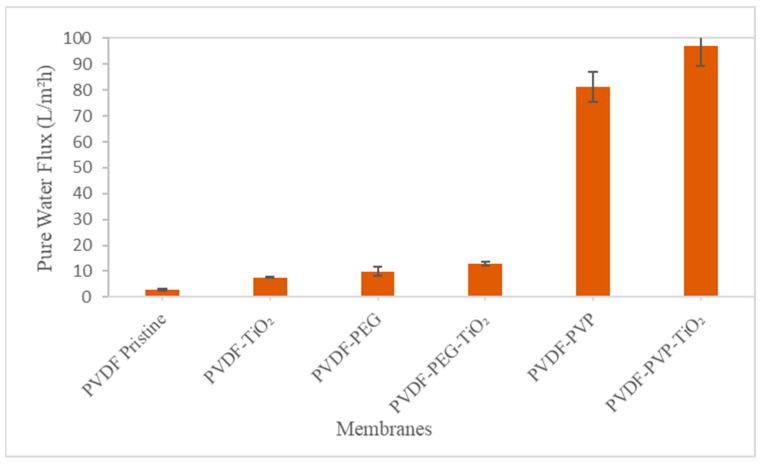
Pure water flux of membranes.

**Figure 7 nanomaterials-13-01023-f007:**
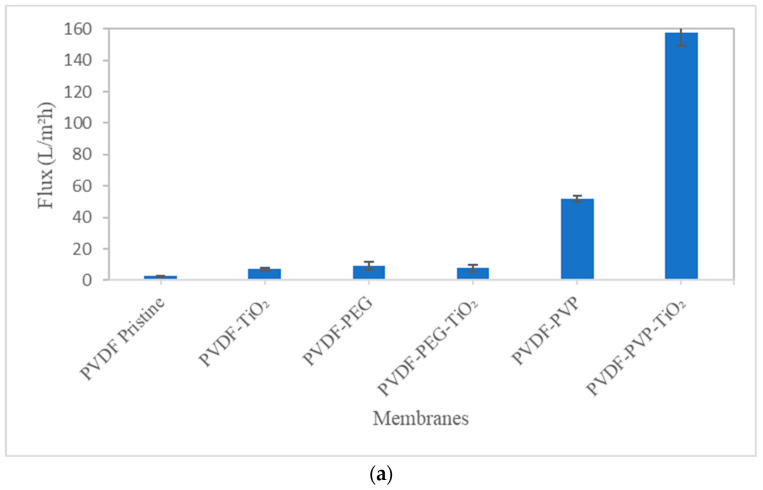
(**a**) BSA solution flux of membranes. (**b**) BSA rejection of membranes.

**Table 1 nanomaterials-13-01023-t001:** Dope solutions for flat sheet membranes.

Membrane	PVDF (%.wt)	TiO_2_ (%.wt)	PVP (%.wt)	PEG (%.wt)	DMAc (%.wt)
Pristine PVDF	16	0	0	0	84.000
PVDF–TiO_2_	16	0.975	0	0	83.025
PVDF–PEG	16	0	0	2	82.000
PVDF–PEG–TiO_2_	16	0.975	0	2	81.025
PVDF–PVP	16	0	2	0	82.000
PVDF–PVP–TiO_2_	16	0.975	2	0	81.025

**Table 2 nanomaterials-13-01023-t002:** Pore size of membranes.

Membrane	Pore Size (nm)
Pristine PVDF	91.3
PVDF–TiO_2_	142
PVDF–PEG	111
PVDF–PEG–TiO_2_	98.0
PVDF–PVP	118
PVDF–PVP–TiO_2_	148

**Table 3 nanomaterials-13-01023-t003:** Porosity of membranes.

Membrane	Porosity (%)
Pristine PVDF	46.493
PVDF–TiO_2_	68.263
PVDF–PEG	78.055
PVDF–PEG–TiO_2_	80.389
PVDF–PVP	88.282
PVDF–PVP–TiO_2_	85.218

**Table 4 nanomaterials-13-01023-t004:** Tensile test of membranes.

Membrane	Tensile Strength (Kpa)	Elongation at Break (%)	Elastic Modulus (Mpa)
Pristine PVDF	2.07	32.31	32.37
PVDF–TiO_2_	1.15	13.39	18.46
PVDF–PEG	3.65	11.25	8.60
PVDF–PEG–TiO_2_	4.06	30.84	28.95
PVDF–PVP	1.55	4.83	16.43
PVDF–PVP–TiO_2_	0.97	19.99	4.47

**Table 5 nanomaterials-13-01023-t005:** Viscosity of membranes’ dope solutions.

Membrane	Viscosity (MPa.s)
Pristine PVDF	1577.5
PVDF–TiO_2_	1789.4
PVDF–PEG	2007.6
PVDF–PEG–TiO_2_	1719.8
PVDF–PVP	2917.0
PVDF–PVP–TiO_2_	1992.5

**Table 6 nanomaterials-13-01023-t006:** Mixed-Matrix Membranes Water Flux Comparison in This Work and Previous Studies.

Membrane	Water Flux (L/m^2^h)	References
PVDF–TiO_2_–PVP	97.01	Present Study
PVDF–TiO_2_–PEG	12.79	Present Study
PVDF–TiO_2_	7.56	Present Study
PVDF–TiO_2_	45.36	[52]
PVDF–TiO_2_	58.81 ± 1.96	[8]
PVDF–TiO_2_	105.1	[34]
PVDF–TiO_2_–PEG	65.74 ± 1.77	[18]
PVDF–TiO_2_–PVP	72.2	[37]
PVDF–TiO_2_–PVP	223	[64]
PVDF/PVDFg–PVP/TiO_2_	41.89	[65]

## Data Availability

The data presented in this work are available on request from the corresponding authors.

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
