# Peer review of "Preparation and Characterization of PVDF–TiO2 Mixed-Matrix Membrane with PVP and PEG as Pore-Forming Agents for BSA Rejection"

_nanomaterials, 2023, doi:10.3390/nano13061023_

Round 1

Reviewer 1 Report

1.         The molecular weights of PVP and PEG should be provided in the manuscript.

2.         The particle size distribution of TiO2 should be provided in the manuscript.

3.         The TiO2 particles should be indicated in the SEM (figure 2), along with the EDS analysis of the indicated TiO2 regions.

4.         It is not clear that the adsorption of PVP and PEG is due to physical adsorption or chemical adsorption (Lines 195-213). This should be addressed (such as based on FTIR study). The reviewer suggested that additional experiment on the PVP and PEG effect on the particle size distribution of TiO2 should be conducted, like laser size analysis.

5.         Lines 200-201. “Introduction of PEG to PVDF membranes reduced the casting solution's thermodynamic stability”. This statement was not clear. The PEG was adsorbed on to the surface of TiO2. My opinion is that the distribution of PEG PVP modified TiO2 will affect the pore structure of PVDF.

6.         Additional cyclic filtration experiments should be conducted

7.         Since PEG was used as dispersant of TiO2 nanoparticle, the statement in lines 296-298, i.e. “The aggregation of TiO2 nanoparticles in the PVDF-TiO2-PVP (82.568o) and PVDF-TiO2-PEG (80.046o) membranes led to a greater contact angle than the PVDF-TiO2 membrane.”, may not be correct.

8.         Lines 309-312. “Previous studies found that incorporating 0.156% TiO2 nanoparticles into composite membranes lowered contact angles by 2.8 percent PEG was used as a dispersant of TiO2 nanoparticles before the casting solution due to its steric hindrance effects [49].” The statement was not clear.

9.         Lines 33-341. “Adding 2% PEG to the PVDF-TiO2 membrane increased the percentage of surface porosity from 68% to 80%, proving that an increment in PEG content improves surface porosity as shown in Figure 6.” In the manuscript, Figure 6 was pure water flux of membrane. I thought Figure 6 was misplaced or missing.

10.     The reviewer suggested that different molecular weight of PVP and PEG should be investigated, since both molecular weight and content of PVP and PEG will affect the viscosity, which in turn many affect the morphology of the pore structure of the membrane and subsequent transport performances the membrane.

11.     Lines 438-439. “The PVP blending shift is not permanent, a portion of the PVP gets washed away after use and PVP functions as a pore-forming agent”. Additional experiment on the removal of PVP should be conducted.

12. The reviewer suggested that the desorption of BAS should be conducted in order to evaluate the ability of regeneration of the membrane.

Author Response

Dear reviewer, Please see the attachment to see the responses.

Reviewer 2 Report

The topic of modifying membranes made from PVDF by adding TiO2 and PEG or PVP is well known and well described in the literature (e.g. J.-P.Méricq, J.Mendret, S.Brosillon,C.Faur, High performance PVDF-TiO2 mebranes for water treatment. Chemical Engineering Science 123 (2015) 283–291; Song, H.,Shao,J.,He,Y.,Liu,B.,Zhong,X.,2012.Natural organic matter removal and flux decline with PEG–TiO2-doped PVDF. J.Membr.Sci.405-406,48–56).

The reviewed work does not contain new elements, and also contains some errors in the interpretation of the obtained results, which is described in detail in the attached file.

Author Response

(The authors gave the same response as above.)

Reviewer 3 Report

Comments to the Authors

In this manuscript authors studied the performance of PVDF/TiO2 mixed-matrix membranes (MMMs) with pore-forming agents like PVP and PEG. This research has value for the researchers in the related areas. However, the paper needs improvement before acceptance for publication. My detailed comments are as follow:

1.      In the introduction section authors should introduced following interesting articles related to PVDF:

a.       doi.org/10.1021/acs.iecr.0c03069

b.      doi.org/10.1016/j.nanoso.2020.100487

2.      Authors should mark the size of TiO2 particles in the microscopic images.

3.      The quality of the figure 4 should be improved and the characteristic peaks mentioned in the figure.

4.      There are lots of typos errors like “PVDF/PVP/TiO2 AND PVDF/PEG/TiO2”

5.      The objective writing of the manuscript should be improved.

6.      Authors should include future fate of such research.       

Author Response

(The authors gave the same response as above.)

Reviewer 4 Report

This manuscript investigated the “ Preparation and Characterization of PVDF-TiO2 Mixed Matrix Membrane with PVP and PEG as Pore Forming Agents for BSA Rejection. The subject of this protein rejection is interesting using MMMs, but the novelty of present work is lacking when compared with existing literature. Therefore, the current manuscript is not suitable for publication in Nanomaterials Journal. The list of commends are shown below.

1.     The 1st two sentence of abstract (line 19-20) and 2nd paragraph of introduction (line 49-50) shows similar meaning of writing and shows repetition.

2.     In abstract author stated as “Increases in hydrophilicity, viscosity, flux in pure water, and BSA solution were achieved using PVP and PEG additives”. What is the purpose and role of TiO2 incorporation in the composite membrane?

3.     In abstract and conclusion, add the novelty of the present work.

4.     The author only mentioned general properties of PVDF and TiO2 in introduction section. There is no specific MMM performance which related to present work. Many sentences don’t have continuous flow and difficult to understand. The introduction section needs to revise completely.

5.     “….has been carried out in this study to address the problem above.” What type of problems are addressed and detailed clearly? I didn’t see this in introduction section.  

6.     “the membrane was soaked in tap water” Why using tap water instead of DI water?

7.     Why the author used 0.975 and 2% addition of TiO2 and PVP/PEG in the composite membrane. What is the optimization level?

8.     Many grammatical and typo errors are found in the entire manuscript. For example, “Contact Angle Analysis”, “PVDF/PVP/TiO2 AND PVDF/PEG/TiO2, “modification transforms membranes”, “16 %wt” etc.

9.     The author mention 91-148 nm of pore size using SEM analysis. However, the Fig. 3 shows microsized pores? The pore size is very important to determine both permeance and rejection efficiency.

10.  Figure 4 shows no difference as compared with pure PVDF? Moreover, the author mentioned peak position in the writing rather than comparision.

11.  The author should also add the surface charge of the membranes.

12.  The flux value is lower as compared with many other literature reports. The author should compare and highlight the discussion of present work.

13.  Higher flux shows lower BSA rejections or vice-versa, particularly TiO2/PEG or TiO2/PVP sample. Why? The detail discussion is missing in the manuscript. What is the purpose of using TiO2 addition in this work if the PEG or PVP already have the improving trends of ultrafiltration (as compared with pure PVDF)?

14.  English language and grammar needs improvement.  

Author Response

(The authors gave the same response as above.)

Reviewer 5 Report

A manuscript titled “Preparation and characterization of PVDF-TiO2 mixed matrix membrane with PVP and PEG as pore forming agents for BSA rejection” by Rianyza Gayatri et al. This manuscript explained the fabrication of PVDF-based membranes for BSA rejection. This work is acceptable; however, some minor issues should be addressed before published in “Nanomaterials”. This article can be improved by addressing the following issues:

11.      The authors should follow the same text format throughout the manuscript.

22.      What is the thickness of fabricated membranes?

33.      Why did the authors choose PVDF for BSA rejection?

44.      Should the authors clearly state the novelty of his work?

55.      The authors should attach the EDX report of his fabricated membrane.

66.      Figure 4. FTIR spectra of membranes, should the author indicate each peak ranges.

77. Some of the important recent references based on the application of the membrane need to cite in a suitable place in the revised manuscript for better-providing information. Examples: doi.org/10.1002/er.7477; 10.1016/j.memsci.2021.119435.

Author Response

(The authors gave the same response as above.)

Round 2

Reviewer 2 Report

The manuscript has been sufficiently improved

Author Response

Thanks to the reviewers for the  insightful comments. The manuscript's quality has improved due to the invaluable comments and suggestions provided by reviewers.

Reviewer 4 Report

The author revised the manuscript accordingly and improve its quality of the manuscript. The current manuscript is suitable for publication in Nanomaterials Journal after considering minor revisions.

1. Based on the previous Q12 of lower flux values, the author intends to compare lower values of flux data. the author should also compare other recent literature of higher value and highlight the concerns or advantages of present work data. In addition, the author may add a comparison table for easy understanding for the readers.

Author Response

Dear reviewer, thanks for the invaluable comments. Please see the attachment to see the responses.
